# Prevalence of Ranavirus Infection in Three Anuran Species across South Korea

**DOI:** 10.3390/v14051073

**Published:** 2022-05-17

**Authors:** Namho Roh, Jaejin Park, Jongsun Kim, Hyerim Kwon, Daesik Park

**Affiliations:** 1Department of Biological Sciences, Kangwon National University, Chuncheon 24341, Kangwon, Korea; shskagh@naver.com; 2Division of Science Education, Kangwon National University, Chuncheon 24341, Kangwon, Korea; zhqnfth1217@naver.com (J.P.); jongsun331@naver.com (J.K.); aquila_8@naver.com (H.K.)

**Keywords:** amphibian, agricultural areas, infectious disease, introduced species

## Abstract

To cope with amphibian die-offs caused by ranavirus, it is important to know the underlying ranavirus prevalence in a region. We studied the ranavirus prevalence in tadpoles of two native and one introduced anuran species inhabiting agricultural and surrounding areas at 49 locations across eight provinces of South Korea by applying qPCR. The local ranavirus prevalence and the individual infection rates at infected locations were 32.6% and 16.1%, respectively, for *Dryophytes japonicus* (Japanese tree frog); 25.6% and 26.1% for *Pelophylax nigromaculatus* (Black-spotted pond frog); and 30.5% and 50.0% for *Lithobates catesbeianus* (American bullfrog). The individual infection rate of *L. catesbeianus* was significantly greater than that of *D. japonicus*. The individual infection rate of *P. nigromaculatus* was related to the site-specific precipitation and air temperature. The individual infection rate gradually increased from Gosner development stage 39, and intermittent infection was confirmed in the early and middle developmental stages. Our results show that ranavirus is widespread among wild amphibians living in agricultural areas of South Korea, and mass die-offs by ranavirus could occur at any time.

## 1. Introduction

Ranavirus is an infectious agent that affects ectothermic organisms such as fish, amphibians, and reptiles [1,2,3]. Ranavirus is known as one of the main causes of the global decline in amphibians, in addition to chytrid fungi [4,5,6,7]. As a major threat to various amphibians, ranavirus, for example, has caused mass die-offs of the endangered giant salamander in China [8], the larval group of endangered gold-spotted pond frogs in Korea [9], and the abundance of common frogs in England [10]. Despite being a continuous threat, ranaviruses are much less well studied than the amphibian chytrid fungus [11,12,13]. Since ranavirus was first discovered in the United States [14], it has spread and is now found on five continents [3,7,15]. Recently, the number of cases of discovery and mass die-offs due to ranavirus infection has also increased in Asia.

Among Asian countries, China has had the highest number of published studies, with 153 registered publications in the Web of Science database between 2010 and 2019 [10]. In addition to China, cases of ranavirus infection or mass die-offs due to ranavirus infection have been reported in eight Asian countries and regions, including Taiwan, Hong Kong, Japan, Korea, Thailand, Singapore, Malaysia, and India [16]. To date, ranavirus infection or mass mortality reports have been made for five amphibian species in South Korea, including larval *Rana huanrenensis* [17], adult *R. uenoi* [18], adult *Kaloula borealis* [19], larval *P. chosenicus* [12], and larval *D. japonicus* [19]. To predict the possibility of ranavirus-caused mass die-offs according to species and region, it is important to determine the ranavirus prevalence in various amphibian species in a wide area. While ranavirus-caused mass die-offs have been frequently reported in various countries [20,21], few studies have screened ranavirus prevalence on a regional scale [21,22,23]. Even within China, where the most research has been performed, screening was performed only in Heilongjiang Province and some areas of northeastern China, and the studies included only a few species, such as *R. dybowskii* and *R amurensis* [24,25]. Such background studies have not been conducted in most Asian countries.

To determine the ranavirus prevalence in amphibian species inhabiting agricultural and surrounding areas, such as wetlands, ponds, and reservoirs, we screened the ranavirus prevalence in three anuran species: the introduced *L. catesbeianus* and the native *D. japonicus* and *P. nigromaculatus*. Our study covered 49 locations across eight provinces in South Korea. In addition, the relationship between the individual ranavirus infection rate and habitat characteristics was explored.

## 2. Materials and Methods

### 2.1. Sampling Locations and Tissue Collection

The selected three species (*D. japonicus*, *P. nigromaculatus*, and *L. catesbeianus*) are commonly distributed in agricultural areas in South Korea [26]. We selected *L. catesbeianus* because it is an introduced species widely inhabiting the central and southern parts of the Korean Peninsula [27], and there have been many reports of ranavirus infection in this species in many countries [28,29,30]. *Dryophytes japonicus* and *P. nigromaculatus* largely share distribution ranges and habitats with *L. catesbeianus*, and they are also representative anuran species in rice paddies [26]. To identify sampling locations, we first selected the areas where all three species or at least two native species have been reported within a 3 km radius based on the results of the third (2006–2012) and fourth (2013–2018) “National Ecosystem Surveys (NESs)”. The South Korean Ministry of Environment has conducted three NESs since 1986, and the data are available from EcoBank [31]. After the confirmation of species presence by directly visiting the locations, we finally selected 49 locations across eight provinces (Figure 1; Table 1). Considering the effective collection of samples over a wide range, we collected tadpoles of the three species using a dip net, seine or trap between June 2020 and October 2021. To prevent potential transmission of ranavirus between sampling locations, we sanitized all collecting equipment using the following protocol: brushing off mud and vegetation, spraying 10% bleach (Yuhan Clorox, Seoul, South Korea) and cleaning with tap water, spraying 70% ETOH, and completely air-drying all field equipment after each collection [32]. We individually preserved sampled tadpoles (Voucher number G01621DJ-G03995LC) in 99% ETOH after euthanasia by submerging them for more than 15 min in 0.5% MS222 [33]. In the laboratory, we randomly subsampled a maximum of eight tadpoles of each species from each location, determined their Gosner developmental stage [34], extracted liver tissues, and preserved the tissues at −80 °C.

### 2.2. DNA Extraction and qPCR

DNA extraction from liver tissue was performed with the DNeasy Blood and Tissue kit (Qiagen, Hilden, Germany) following the manufacturer’s protocol. The extracted DNA was quantified with a Qubit3 Fluorometer (Invitrogen, Waltham, MA, USA) using the Qubit 1X dsDNA HS Assay Kit (Invitrogen, Waltham, MA, USA) and then stored at −80 °C until qPCR experiments. We used RVMCPKim3_F and RVMCPKim3_R qPCR primers to determine ranavirus infection [35]. The composition of the amplification reaction solution for qPCR included 10 µL of power SYBR green PCR master mix (Applied Biosystems, Waltham, MA, USA), 0.5 µL of forward primer, 0.5 µL of reverse primer, and 4 ng of DNA. We adjusted the volume to a final value of 20 µL using molecular biology grade water. qPCR was performed in QuantStudio 1 (Applied Biosystems, Waltham, MA, USA) at 95 °C for 10 min, followed by 40 cycles of 95 °C for 15 s and 62.5 °C for 20 s. qPCR of all samples was performed in triplicate with a negative control (sterile, molecular grade water) and a positive control. The positive control was PCR-amplified ranavirus MCP DNA from *Kaloula borealis*, and this sample was obtained and verified for ranavirus infection in 2016 [19]. If a positive reaction was detected in two or more wells of the three replicate samples, the melting temperature (Tm) value of the melting curve coincided with the positive control, and the cycle threshold (CT) value was 35 or less, then we considered the tadpole to be infected with ranavirus (Appendix A). If a positive reaction was confirmed in only one well, the test was rerun, and only when the above conditions were satisfied was it judged to be an infected tadpole.

### 2.3. Weather Factors and Habitat Characteristics

To investigate the relationship between the ranavirus infection rate and habitat characteristics, we analyzed 16 habitat characteristics at each sampling location. The land cover rate (urban area, agricultural area, forest area, grass area, wetland area, bare area, hydrosphere area) within a radius of 3 km^2^ from the mid-sampling point was calculated on the 2020 South Korean land cover map [36]. From the land cover data, greenhouse cultivation areas within agricultural areas, salt fields and tidal flat areas within wetland areas, and sea areas within hydrosphere areas were excluded during data handling because these areas are inhabitable for amphibians. Additionally, the shortest distance from the mid-sampling point to agricultural land, mountains, water bodies, and urban areas was calculated in units of 1 m. When a sampling point was located within a specific coverage area, the distance was set to 2 m. All land cover rates, distance data, and altitude values of each sampling point were calculated using QGIS (ver. 3.4.7, QGIS.org 2021; https://www.qgis.org/ko, accessed on 27 July 2021). Furthermore, the average air temperature, lowest air temperature, highest air temperature, and average precipitation for the immediately preceding quarter, based on the sampling date of the tadpoles used in the qPCR experiment, were obtained from the local meteorological station closest to the sampling point.

### 2.4. Data Analyses

For analyses, we log-transformed 16 habitat and climate characteristics. As most of the data did not present a normal distribution even after transformation, nonparametric statistics were used for statistical analyses. We used the chi-square test to determine the difference in the prevalence of infected locations (local ranavirus prevalence) among the three species using the Kruskal–Wallis test, and the Dunn-Bonferroni post hoc test was conducted following the provided option of the test. The correlations of the individual ranavirus infection rates within the three species and between the three species and the 16 habitat characteristics were verified by Spearman correlation analysis. All statistics were performed in SPSS (Version 26, IBM). To determine the change in the individual ranavirus infection rate according to the Gosner developmental stage, the number of ranavirus-infected individuals was calculated at each stage and then compared to the number of individuals, which was qPCR-tested at each stage for each species. Afterward, the average ranavirus infection rate of the three species based on each developmental stage was obtained, and trends were presented. All data in the text are presented as the mean ± 1 standard error unless otherwise noted.

## 3. Results

Out of 49 populations, we collected larval *D. japonicus* from 43 locations (22.1 ± 4.2 individuals per location, *n* = 43), larval *P. nigromaculatus* from 43 locations (21.0 ± 4.0, *n* = 43), and larval *L. catesbeianus* from 26 locations (19.5 ± 10.2, *n* = 26) (Figure 1). The Gosner developmental stage of the collected larvae ranged from 25 to 46. In the qPCR experiments, we subsampled and used 344 larval *D. japonicus* (8 tadpoles per location), 342 larval *P. nigromaculatus* (8 per location, except for 6 in one location), and 187 larval *L. catesbeianus* (7.2 ± 2.1 SD per location, ranging from 1–8 individuals).

### 3.1. Rates of Local and Individual Ranavirus Infections

We found at least one ranavirus-infected individual at 28 of the 49 studied locations (57.1%) (Figure 1; Table 1). The local ranavirus prevalence was 32.6% (14 out of 43 locations), 25.6% (11 out of 43 locations), and 30.8% (8 out of 26 locations), and the individual infection rates within the infected locations were 16.1 ± 2.0% (12.5–37.5%, *n* = 14), 26.1 ± 5.7% (12.5–75.0%, *n* = 11), and 50.0% 50.0 ± 9.7% (12.5–100%, *n* = 8) for *D. japonicus*, *P. nigromaculatus*, and *L. catesbeianus*, respectively (Table 1). The local ranavirus prevalences were not significantly different between the three species (*p* = 0.77), but the individual ranavirus infection rates at the infected locations were significantly different (H (2) = 12.47, *p* = 0.002) (Figure 2). In particular, the rate of *L. catesbeianus* was greater than that of *D. japonicus* (*p* = 0.001). In contrast, the rates were not significantly different between *L. catesbeianus* and *P. nigromaculatus* or between *P. nigromaculatus* and *D. japonicus* (*p*s > 0.05).

### 3.2. Relationship with Weather Conditions, Habitat Characteristics, and Developmental Stages

The individual infection rate was positively correlated with grass area (*r* = 0.579, *n* = 14, *p* = 0.030) in *D. japonicus* and negatively correlated with distance to mountains (*r* = −0.604, *n* = 11, *p* = 0.049) in *P. nigromaculatus* and with forest area (*r* = −0.781, *n* = 8, *p* = 0.022) in *L. catesbeianus*. In *P. nigromaculatus*, the rate was also positively related to the precipitation (*r* = 0.634, *n* = 11, *p* = 0.036), average air temperature (*r* = 0.837, *n* = 11, *p* = 0.001), and lowest air temperature (*r* = 0.721, *n* = 11, *p* = 0.012) in the first quarter prior to the sampling date. The individual ranavirus infection rates of the three species were not related (*p*s > 0.05). Additionally, all other remaining relationships were not significant (*p*s > 0.05). The individual infection rate of the three species showed a gradual increase from Gosner developmental stage 39. In the early and middle developmental stages, there were intermittent cases of ranavirus infection, such as at stages 32 and 37 (Figure 3).

## 4. Discussion

Our study shows that ranavirus is widely spread in amphibians inhabiting agricultural and surrounding areas in South Korea. Ranavirus-infected tadpoles were identified in over 25.6% of the surveyed locations, and the average individual infection rate in the infected area was as high as 50% depending on the species. Amphibian larvae often have a higher rate of ranavirus infection than adults [37] due to factors such as immunologically vulnerable metamorphosis and high population density [38,39]. To date, although ranavirus infection in larval *D. japonicus* has been confirmed in agricultural areas [19], there has been no report of mass die-offs of either larval or adult *D. japonicus* or *P. nigromaculatus* in South Korea. Our results show that given the high 50% individual infection rate of *L. catesbeianus*, amphibian mass die-offs due to ranavirus could occur in agricultural areas whenever outbreak conditions are present.

The high individual infection rate of *L. catesbeianus* tadpoles may be due to the habitat and ecological characteristics of the species. Ranavirus can infect various ectothermic vertebrates through direct individual contact or indirect water matrices [40,41,42]. In this study, the individual infection rate of *L. catesbeianus* was 50.0%, which was higher than the rates of 16.1% and 26.1% for *D. japonicus* and *P. nigromaculatus*, respectively. Several factors might have affected this result. First, while the larvae of *D. japonicus* and *P. nigromaculatus* generally spend approximately one month in water before metamorphosis [26], the larvae of *L. catesbeianus* often spend more than one year in water [43]. This difference may increase the likelihood of being directly or indirectly infected by ranavirus. Second, considering that the main habitats of *L. catesbeianus* are ponds and reservoirs adjacent to farmland and villages, there is a high possibility of long-term exposure to anthropogenic pollutants. Such pollutants are often related to ranavirus infection [44,45]. Third, adult *L. catesbeianus* live in ponds throughout the year. In contrast, adult *D. japonicus* and *P. nigromaculatus* live in water only during the breeding season [26]. Therefore, adult *L. catesbeianus* are highly likely to be infected with ranavirus, which may have subsequently caused the high rate of larval infection. In future studies, screening the degree of ranavirus infection in adult *L. catesbeianus* is necessary.

In this study, we could not find any association of ranavirus infection between introduced *L. catesbeianus* and two native species (*D. japonicus* and *P. nigromaculatus*). Despite screening 49 locations, we confirmed only four locations with more than two species infected by ranavirus. It has been reported that *L. catesbeianus* functions as a ranavirus carrier and transmission vector [46,47]. *Lithobates catesbeianus* was introduced to Korea approximately 50 years ago and is widely distributed in the central and southern regions of South Korea [27]. In addition, considering that ranavirus has been previously detected in bullfrogs in many countries [28,29,30], *L. catesbeianus* may be responsible, at least in part, for the wide ranavirus prevalence in South Korea. However, in this study, joint ranavirus infections of both *L. catesbeianus* and *P. nigromaculatus* were confirmed at only one location. Therefore, we could not appropriately test the hypothesis that *L. catesbeianus* may serve as a ranavirus vector for native anuran species. In future studies, screening for ranavirus and comparing the strains through sequencing in amphibians, fish, and insects that share a water source with or cohabit with *L. catesbeianus* are necessary.

The relationships between the individual ranavirus infection rate and several habitat characteristics were identified. In contrast with the cases in *L. catesbeianus*, the infection rate of *P. nigromaculatus* showed a high positive correlation with precipitation, average air temperature, and lowest air temperature in the first quarter prior to the collection date. This result was consistent with those of previous studies showing that the rate of ranavirus infection was related to habitat temperature [2]. In addition, in *P. nigromaculatus*, the greater the distance from the mountain was, the higher the infection rate, implying that the infection rate might be low when the anthropogenic factors are low near mountainous or forest areas. However, in *L. catesbeianus*, there was no correlation with climatic factors, and only the more forested areas had a lower infection rate. This result suggests that ranavirus infection in the introduced *L. catesbeianus* might be related to anthropogenic factors such as various pollutants rather than to climatic factors. In *D. japonicus*, the infection rate at 12 of the locations was the same, at 12.5%, so it might be statistically difficult to detect any significant relationships with habitat characteristics. Many amphibians can survive in a state of ranavirus infection [15,46], and many field studies have conducted post hoc mass mortality analysis [17,18,28]. Thus, the direct causal relationship between population habitat characteristics and ranavirus-caused mass mortality has not been clearly established. Further studies are needed to link the rate of infection with habitat characteristics.

There were two trends in the infection rate according to the developmental stage of tadpoles. The individual ranavirus infection rate gradually increased after Gosner developmental stage 39, and intermittent ranavirus infection occurred in the early to mid-developmental stages. Among the factors influencing the ranavirus infection rate in amphibians, the developmental stage is a well-known factor [2,3,4,5]. The ranavirus infection rate is often high at Gosner stages 44–46, which is when metamorphosis occurs and often leads to die-off events [38,48,49]. There was also a case report in South Korea where many larval *R. huanrenensis* died during metamorphosis [17]. Although ranavirus-caused die-off events of amphibian tadpoles are often found close to metamorphosis [50,51], tadpoles at lower developmental stages are also susceptible to ranavirus infection [52]. Our results imply that anuran tadpoles can be infected by ranavirus regardless of the specific developmental stage and that ranavirus-caused mass die-offs can occur whenever environmental triggers form suitable conditions for an outbreak. In the paddy fields in which rice is cultivated, water flooding and draining are repeated to help the successful growth of rice, and this cycle greatly affects amphibian activities [39,53]. Often, several thousand dead tadpoles at various developmental stages are found in small hollow patches in water-drained rice paddies in summer [54]. Our findings suggest a potential link between these deaths and ranavirus, suggesting the urgent need for further studies.

## 5. Conclusions

Our study is of great significance in that we, for the first time, determined the ranavirus prevalence in representative amphibian species living in agricultural and surrounding areas across South Korea. Our results show that mass die-offs of amphibians due to ranavirus could occur at any time in such areas. In particular, the further screening of adult frogs and the investigation of possible ranavirus causation on large die-offs of tadpoles in rice fields are urgently necessary.

## Figures and Tables

**Figure 1 viruses-14-01073-f001:**
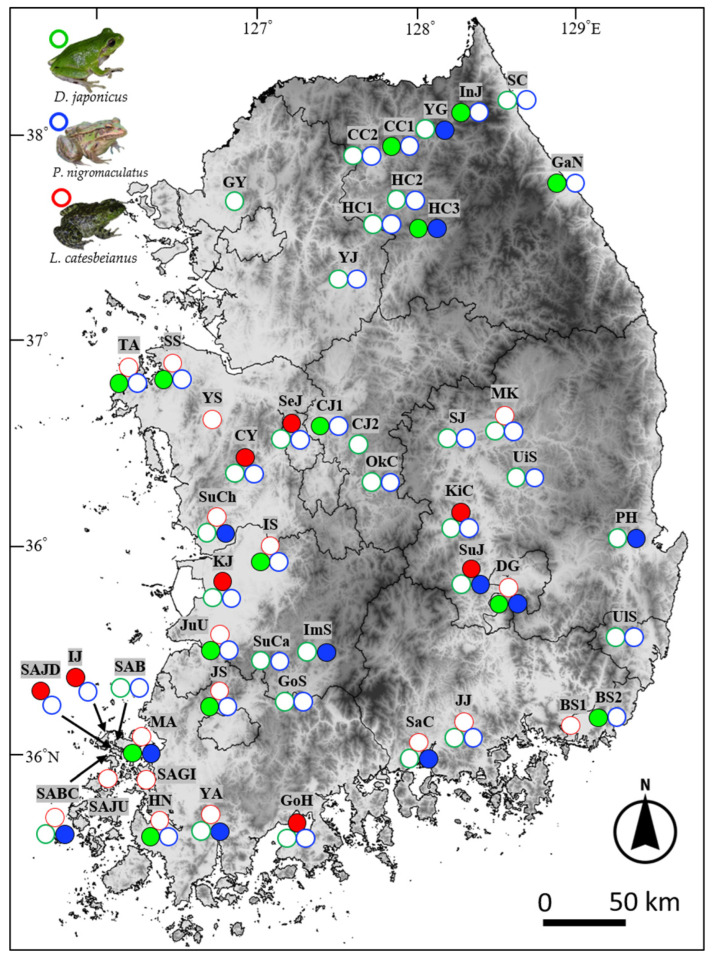
Local ranavirus infection in larvae of three anuran species (*Dryophytes japonicus*, green circles; *Pelophylax nigromaculatus*, blue circles; *Lithobates catesbeianus*, red circles) at 49 locations across eight provinces in South Korea. Filled circles indicate the confirmation of at least one ranavirus-infected tadpole of the species at the location.

**Figure 2 viruses-14-01073-f002:**
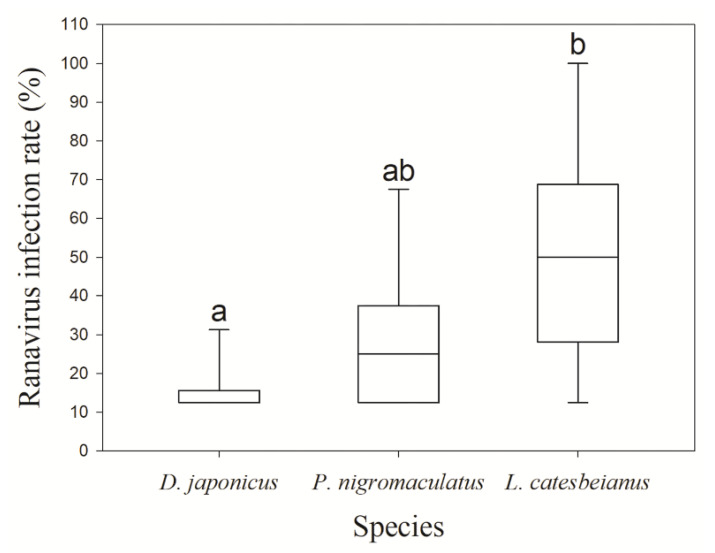
Box plots (medians ± quartiles) of the ranavirus infection rates among larvae of three anuran species (*Dryophytes japonicus*, *Pelophylax nigromaculatus*, *Lithobates catesbeianus*). The different characters on the bars between species indicate significantly different rates using the Kruskal–Wallis test with the Dunn-Bonferroni post hoc test (*p* = 0.001).

**Figure 3 viruses-14-01073-f003:**
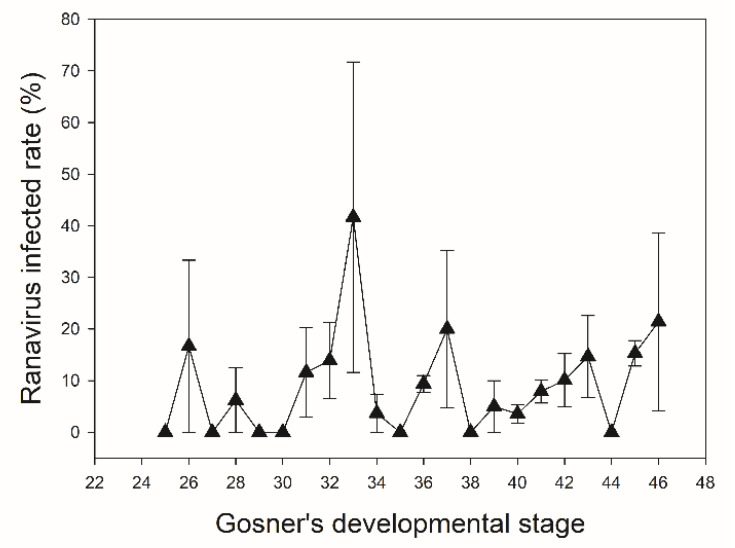
Changes in the ranavirus infection rate according to the Gosner developmental stage, based on combined data from larvae of three anuran species (*Dryophytes japonicus*, *Pelophylax nigromaculatus*, *Lithobates catesbeianus*).

**Table 1 viruses-14-01073-t001:** Sampling locations and ranavirus infection rates in larvae of three anuran species (*Dryophytes japonicus*, *Pelophylax nigromaculatus*, *Lithobates catesbeianus*) across 49 sampling locations in South Korea.

Province	Sampling Location (Abbreviation)	Infection Rate (%) (No. of Infected/Tested Tadpoles)		
		*D. japonicus*	*P. nigromaculatus*	*L. catesbeianus*
Gangwon	Sokcho (SC)	None	None	-
	Gangneung (GaN)	12.5 (1/8)	None	-
	Inje (InJ)	12.5 (1/8)	None	-
	Yanggu (YG)	None	12.5 (1/8)	-
	Chuncheon1 (CC1)	12.5 (1/8)	None	-
	Chuncheon2 (CC2)	None	None	-
	Hongcheon1 (HC1)	None	None	-
	Hongcheon2 (HC2)	None	None	-
	Hongcheon3 (HC3)	25.0 (2/8)	75.0 (6/8)	-
Gyeonggi	Yeoju (YJ)	None	None	-
	Goyang (GY)	None	-	-
Chungbuk	Cheongju1 (CJ1)	12.5 (1/8)	None	-
	Cheongju2 (CJ2)	None	-	-
	Okcheon (OkC)	None	None	-
Chungnam	Yesan (YS)	-	-	None
	Seosan (SS)	12.5 (1/8)	None	None
	Taean (TA)	37.5 (3/8)	None	None
	Sejong (SeJ)	None	None	50.0 (1/2)
	Cheongyang (CY)	None	None	25.0 (2/8)
	Seocheon (SuCh)	None	25.0 (2/8)	None
Gyeongbuk	Mungyeong (MK)	None	None	None
	Sangju (SJ)	None	None	-
	Uisung (UiS)	None	None	-
	Gimcheon (KiC)	None	None	100.0 (1/1)
	Seongju (SuJ)	None	12.5 (1/8)	12.5 (1/8)
	Daegu (DG)	12.5 (1/8)	12.5 (1/8)	None
	Pohang (PH)	None	37.5 (3/8)	-
Gyeongnam	Ulsan (UlS)	None	None	-
	Busan1 (BS1)	-	-	None
	Busan2 (BS2)	12.5 (1/8)	None	-
	Sacheon (SaC)	None	37.5 (3/8)	None
	Jinju (JJ)	None	None	None
Jeonbuk	Iksan (IS)	12.5 (1/8)	None	None
	Gimjae (KJ)	None	None	75.0 (6/8)
	Jeongup (JuU)	12.5 (1/8)	None	None
	Imsil (ImS)	None	12.5 (1/8)	-
	Sunchang (SuCa)	None	None	-
Jeonnam	Jangseong (JS)	12.5 (1/8)	None	None
	Gokseong (GoS)	None	None	-
	Muan (MA)	12.5 (1/8)	25.0 (2/8)	None
	Imja (IJ)	-	None	50.0 (4/8)
	Sinan1 (SAB)	None	None	-
	Sinan2 (SABC)	None	12.5 (1/8)	None
	Sinan3 (SAGI)	-	-	None
	Sinan4 (SAJD)	-	None	37.5 (3/8)
	Sinan5 (SAJU)	-	-	None
	Yeongam (YA)	None	25.0 (2/8)	None
	Haenam (HN)	25.0 (2/8)	None	None
	Goheung (GoH)	None	None	50.0 (4/8)
Average (No. of infected/tested tadpoles)	49 locations	16.1% (18/344)	26.1% (23/342)	50.0% (22/187)

## Data Availability

Data on the weather and habitat characteristics, collected from 49 sampling sites and the representative qPCR results and the representative photographs of the gel electrophoresis of MCP fragment (~520 bp), which were taken during further sequencing procedures, are openly available in Dryad Digital Repository at https://datadryad.org/stash/dataset/doi:10.5061/dryad.0zpc8670q (accessed on 11 May 2021).

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
