# Peer review of "Prevalence of Ranavirus Infection in Three Anuran Species across South Korea"

_viruses, 2022, doi:10.3390/v14051073_

Round 1

Reviewer 1 Report

Review: Prevalence of ranavirus infection in three anuran species across South Korea

This is a well thought out and a potentially very important paper, especially in Asia, when it comes to ranavirus distributions, prevalence in various species, and country wide surveys.

There are minor issues with the paper that should only require a quick remediation via editing. The methods are sound and the presentation of results is generally very good. I really like the map!

Specific comments:

Line 14 – 17
I found this to be confusing. Please reword so that it is clearer.

Line 27
Should be edited:
Ranavirus is an infectious AGENT (not disease because the disease that is caused by ranavirus infection is termed ranavirosis) that AFFECTS ectothermic…

Line 29 – 31
This sentence is misleading and I would strongly suggest removing it from the introduction.
Ranaviruses are known around the globe to have high morbidity and mortality rates in many populations of amphibians, reptiles and fish. Any statement to the contrary is false, I understand that the authors are trying to get at the point that ranaviruses are much less well studied than the amphibian chytrid fungus, but there are better ways to make this statement.

Line 38
Awkward wording:
Perhaps something like:  Among Asian countries, China HAS HAD THE HIGHEST NUMBER OF PUBLISHED STUDIES, WITH 153 publications…..

Lines 76 – 80
This is not a standard method/protocol for disinfection for ranaviruses between field sites. Please comment on it’s efficacy and provide a reference for it.
There are numerous publications that specify specific disinfection methods for all amphibian diseases (and all herp diseases) that should be followed in the field.

Lines 190-191
The ranges here do not make sense. Please reword the sentences so that the inclusion of ranges is appropriate or remove the ranges.

Line 227
In addition to screening for ranaviruses in all species, it would also be necessary to fully sequence the ranaviruses present in all of the species to ensure that it is the same virus present in all.

Author Response

Response to the Reviewer I

I greatly appreciate the valuable and very useful comments from the Reviewer I, which have helped to improve the manuscript. While revising the manuscript, I carefully considered all the comments. Below, I describe how the comments were incorporated into the revised manuscript. The revised content is indicated in blue color in the manuscript.

Specific comments:

Line 14 – 17: I found this to be confusing. Please reword so that it is clearer.

==> we revised the sentence at L14-17: “... and the individual infection rates at infected locations were 32.6% and 16.1%, respectively, for Dryophytes japonicus (Japanese tree frog); 25.6% and 26.1% for Pelophylax nigromaculatus (Black-spotted pond frog); 30.5% and 50.0% for Lithobates catesbeianus (American bullfrog)”

Line 27: Should be edited: Ranavirus is an infectious AGENT (not disease because the disease that is caused by ranavirus infection is termed ranavirosis) that AFFECTS ectothermic…

==> we incorporated the comment into the sentence at L27: “.. is an infectious agent that affects ectothermic...”

Line 29 – 31: This sentence is misleading and I would strongly suggest removing it from the introduction. Ranaviruses are known around the globe to have high morbidity and mortality rates in many populations of amphibians, reptiles and fish. Any statement to the contrary is false, I understand that the authors are trying to get at the point that ranaviruses are much less well studied than the amphibian chytrid fungus, but there are better ways to make this statement.

==> we revised the sentence following the comments: revised Line 29-30 “ As a major threat to various amphibians, ranavirus, for example, ”and added a sentence at L32-33: “Despite being a continuous threat, ranaviruses are much less well studied than the amphibian chytrid fungus [11-13]”

Line 38: Awkward wording: Perhaps something like:  Among Asian countries, China HAS HAD THE HIGHEST NUMBER OF PUBLISHED STUDIES, WITH 153 publications…..

==> revised L37: “... had the highest number of published studies, with...”

Lines 76 – 80: This is not a standard method/protocol for disinfection for ranaviruses between field sites. Please comment on it’s efficacy and provide a reference for it. There are numerous publications that specify specific disinfection methods for all amphibian diseases (and all herp diseases) that should be followed in the field.

==> we revised the sentence and provided a reference based on the comments at L76-79: “using the following protocol: brushing off mud and vegetation, spraying 10% formalin and cleaning with tap water, spraying 70% ETOH, and completely air-drying all filed equipment after each collection [32]”

Lines 190-191: The ranges here do not make sense. Please reword the sentences so that the inclusion of ranges is appropriate or remove the ranges.

==> We removed and revised the sentence at L190-191: “ in over 25.6% of ... as high as 50% depending on the species”

Line 227: In addition to screening for ranaviruses in all species, it would also be necessary to fully sequence the ranaviruses present in all of the species to ensure that it is the same virus present in all.

==> We revised the sentence at L227-L228: “ and comparing the strains through sequencing...are…”

==> In addition, the manuscript was carefully checked English usage again by two English editing specialists in related study areas. Further improving was indicated in blue colors. I have attached the English editing certificate (4007-1ACD-F116-E4A6-FD6P) of American Journal Expert.

Reviewer 2 Report

Ranavirus is an infectious disease that infects ectothermic organisms such as fish, amphibians, and reptiles.In the present study,  the ranavirus prevalence in tadpoles of two native and one introduced anuran species inhabiting agricultural and surrounding areas at 49 locations across eight provinces of South Korea by applying qPCR was studied.
1. line 39-41,  eight Asian countries should be changed to eight Asian countries and regions.
2.Some anatomical pictures can be reflected in the manuscirpt.
3. It's better to add some histopathological pictures.
4.PCR results from gel electrophoresis are best added.
5. line 160, all other remaining comparisons were not significant (Ps > 0.05),why?
6.There are many errors in grammar and syntax throughout the text of the manuscript, which are required for correction.

Author Response

Response to the Reviewer 2

I greatly appreciate the valuable and very useful comments from the Reviewer 2, which have helped to improve the manuscript. While revising the manuscript, I carefully considered the comments. Below, I describe how the comments were incorporated into the revised manuscript. When I could not directly address the comments, I included an explanation for the reason. The revised content is indicated in blue color in the manuscript.

Specific comments:

1. line 39-41, eight Asian countries should be changed to eight Asian countries and regions.

==> We revised following the comment at L40: “countries and regions”

2. Some anatomical pictures can be reflected in the manuscript.

==> Considering the large number of sample specimens for the background study, we did not dissect the specimens in this study, but focused on qPCR detection of the infection.

3. It's better to add some histopathological pictures.

==> Considering the large number of sample specimens for the background study, we did not conduct histopathological study in this study. During field sampling, we could not notify distinctive pathological symptoms from the samples.

4. PCR results from gel electrophoresis are best added.

==> We provide the figures of representative qPCR results and the photographs of representative gel electrophoresis of the MCP fragment (~520 bp), which were taken during further sequencing procedures, as supplementary materials. We described this addition to the section of the Data Availability Statement (L285-286).

5. line 160, all other remaining comparisons were not significant (Ps > 0.05), why?

==> We revised the sentence to make meaning clearer at L160-161: “In contrast, the rates were not significantly different between P. nigromaculatus and L. catesbeianus or between D. japoncius and P. nigromaculatus

6.There are many errors in grammar and syntax throughout the text of the manuscript, which are required for correction.

==> The manuscript was carefully checked English usage again by two English editing specialists in related study areas. Further improving was indicated in blue colors. The number of the English editing certificate  is 4007-1ACD-F116-E4A6-FD6P, issued by American Journal Expert.

Round 2

Reviewer 2 Report

The author has completed the modification of the manuscript.